# Global economic impacts of COVID-19 lockdown measures stand out in high-frequency shipping data

Jasper Verschuur[1]*, Elco E. Koks[1,2], Jim W. Hall[1]

**1** Environmental Change Institute, University of Oxford, Oxford, United Kingdom, **2** Institute for Environmental Studies, Vrije Universiteit Amsterdam, Amsterdam, Netherlands

* jasper.verschuur@keble.ox.ac.uk

**Data Availability Statement:** All derived datasets used for this analysis are made publicly available at Zenodo: 10.5281/zenodo.4146993. The policy indicators are obtained from the Oxford Coronavirus Government Response Tracker

## Abstract

The implementation of large-scale containment measures by governments to contain the spread of the COVID-19 virus has resulted in large impacts to the global economy. Here, we derive a new high-frequency indicator of economic activity using empirical vessel tracking data, and use it to estimate the global maritime trade losses during the first eight months of the pandemic. We go on to use this high-frequency dataset to infer the effect of individual non-pharmaceutical interventions on maritime exports, which we use as a proxy of economic activity. Our results show widespread port-level trade losses, with the largest absolute losses found for ports in China, the Middle-East and Western Europe, associated with the collapse of specific supply-chains (e.g. oil, vehicle manufacturing). In total, we estimate that global maritime trade reduced by -7.0% to -9.6% during the first eight months of 2020, which is equal to around 206–286 million tonnes in volume losses and up to 225–412 billion USD in value losses. We find large sectoral and geographical disparities in impacts. Manufacturing sectors are hit hardest, with losses up to 11.8%, whilst some small islands developing states and low-income economies suffered the largest relative trade losses. Moreover, we find a clear negative impact of COVID-19 related school and public transport closures on country-wide exports. Overall, we show how real-time indicators of economic activity can inform policy-makers about the impacts of individual policies on the economy, and can support economic recovery efforts by allocating funds to the hardest hit economies and sectors.

## Introduction

The emergence and spread of COVID-19, caused by the severe acute respiratory syndrome coronavirus 2 (SARS-CoV-2), has forced countries worldwide to implement Non-Pharmaceutical Interventions (NPI) to reduce the spread of the virus [1–4]. These NPI, which include among others international travel restrictions, business closures, prohibition of large-scale private and public gatherings, and mandatory quarantines, have shown to effectively reduce the rate of transmission of the virus [1, 3, 5]. As a consequence, however, such policies have had large economic repercussions, both in terms of domestic industry output and international

(https://www.bsg.ox.ac.uk/research/research-projects/coronavirus-government-response-tracker). The UN Comtrade mode of transport data can accessed using the online data portal https://comtrade.un.org. The raw AIS data is provided through the UN global platform. More information can be found on https://unstats.un.org/wiki/display/AIS/.

**Funding:** This research is supported by the University of Oxford COVID-19 Research Response Fund. J.V. acknowledges funding from the Engineering and Physical Sciences Research Council (EPSRC) under grant number EP/R513295/1. E.E.K. was further supported by the Netherlands Organization for Scientific Research NOW (grant no. VI.Veni.194.033).

**Competing interests:** The authors have declared that no competing interests exist.

trade, due to diminishing production and reduced demand for some goods. For instance, model-based estimates show that the global industry value-added may have dropped by 25–40%, depending upon the scale and severity of the implementation of NPI [6].

Quantifying the costs and benefits of various NPI on the economy and global trade is necessary to inform effective policy responses and to navigate the trade-off between slowing the pace of the pandemic and limiting economic impacts [5, 7, 8]. However, monitoring the extent, and understanding the underlying causes, of the economic disruption on a global scale is hard for three reasons; (1) the traditional macroeconomic indicators (e.g. trade, industry output) are often published at several months of delay, (2) their aggregate nature makes it hard to decipher the importance of various impact mechanisms, and (3) macroeconomic indicators are primarily available for high and upper middle-income countries, thereby limiting our ability to understand what is happening in low- and middle-income countries. Therefore, alternative, higher temporal frequency, proxy data of economic activity with a global extent could help improve our understanding of the unfolding economic disruptions to economies globally [5]. Moreover, one could leverage the cross-country heterogeneity in timing and severity of NPI to examine the economic impacts of individual NPI across countries.

A growing body of literature has used high-frequency data (HFD), such as electricity consumption [8, 9], air pollution [5, 10, 11], night-time light intensity [12] and human mobility [13–15], to track the evolution of the pandemic on a country and global scale. In addition, recent research has used HFD sources to quantify the effect of individual NPI on domestic economic output [5, 8, 14]. For instance, Fezzi and Fanghella [9] used daily electricity consumption data for Italy and found that the 3 weeks of severe lockdown reduced the national GDP by almost 30%. Deb et al. [5] used a variety of HFD to estimate the individual impacts of NPI, showing that workplace closures and stay-at-home orders had the largest economic costs. However, proxies such as electricity consumption and human mobility are often hard to relate directly to economic impacts, making it difficult to infer a causal relationship between NPI and economic activity. In addition, these studies often only include countries for which these HFD are available and rarely include countries in the Global South or island nations, making it hard to generalise the results.

In this research, we present a high-frequency dataset of maritime trade flows derived from empirical vessel tracking data, which we use to track the status of global maritime trade during the first eight months of the pandemic. We do this by tracking vessel movements in almost 1200 ports globally in combination with a newly developed algorithm that allows us to estimate trade flows based on this data. Maritime trade flows cover around 80% of the world's trade in terms of volume [16], and can hence be used as a first-order indicator of the status of economic activity in a country and trade between countries. We then apply an econometric model to estimate the impacts of specific NPI on exports by making use of the heterogeneity in the diversity, timing and severity of NPI across countries. The newly derived dataset has a high spatial (166 countries) and temporal (daily) resolution, which helps to track the impacts of the COVID-19 outbreak across a large sample of countries, and provides a more comprehensive picture of changes in economic activity compared to alterative HFD.

We estimate that, globally, maritime trade reduced by 7–9.6% during the first months of 2020, which is equivalent to around 225–412 billion USD. However, large sectoral and geographical disparities are found, with manufacturing sectors being hit the hardest, as well as small island developing states and low-income economies suffering the largest relative losses. We find a negative relationship, robust across different model specifications, between the implementation of COVID-19-related school and public transport closures on country-wide exports. Overall, we provide evidence of how the impacts upon global maritime trade are complex and dependent on the trade-dependencies, sector-composition and NPI implemented.

Our results underline that HFD indicators of economic activity can support governments and international organisations in economic recovery efforts and channelling funds to the hardest hit economies and sectors.

## Method

### Data and trade estimation

We derive estimates of port-level trade flows (imports and exports) for 1153 ports across 166 countries worldwide using the geospatial location and attributes of maritime vessels (from January 2019—August 2020). To do this, we use Automatic Identification System (AIS) data, which provides detailed data on the location, speed, direction and vessel characteristics of all trade-carrying vessels with an AIS transponder (that send information to terrestrial or satellite receivers every few seconds-minutes) [17]. This data is obtained through a partnership with the UN Global Platform AIS Task Team initiative, which aims to develop algorithms and methodologies to make AIS data useful for a variety of fields and applications (traffic, economic trade, fisheries, CO2 emissions).

We develop an algorithm (S1 Appendix) that estimates the trade flows based on the ingoing and outgoing movements of maritime vessels (~3.2 million port calls across 100,000 unique vessels) and their characteristics (e.g. dimensions, utilisation rate, vessel type), going on to disaggregate these trade flows into specific sectors (11 sector classification adopted here). We end up with daily sector-specific trade flow estimates on a port-level, which we aggregate to a country-scale to perform the country-wide impact analysis. This new algorithm significantly advances previous work [18–20] by providing a global scale analysis and being able to provide a sector decomposition.

We validate the results (S1 Appendix) by comparing the derived trade estimates to detailed port-level trade data obtained for five countries (Japan, United Kingdom, United States, New Zealand, Brazil). Moreover, we compare our estimates to country-wide maritime trade flows obtained from UN Comtrade [21] mode of transport data for 27 countries.

### Econometric model

The variation in trade losses across countries are driven by the differences in NPI introduced by countries (in terms of timing, duration, and severity) [5], supply shortages to domestic supply-chains [22], demand reductions in trade-dependent economies [6], and other country specific characteristics (e.g. share of tourism, liberalized credit markets) [23]. NPI can negatively influence industry output by affecting business operations (e.g. workplace closure, mobility restrictions), or positively affect industry output through effectively containing the virus outbreak and thereby allowing industrial production processes and transportation of goods to continue.

To study the implications of NPI on exports (which we use as a proxy of industrial output), we match our daily, country-wide, estimates with data from the Oxford COVID-19 Government Response Tracker (OxCGRT) [24]. Within OxCGRT, data is collected on the implementation and stringency of NPI across 160 countries. We utilise reduced-form econometric techniques [25] to estimate the effect of different containment policies on exports across a balanced sample of 122 countries (for which data is available). The choice of model (a panel regression with fixed effects model) was chosen since the Hausman test [26] showed that omitted country-specific variables are correlated with the explanatory variables (see S2 Appendix). We express export change as the percentage change in detrended exports in 2020 compared to 2019 (S2 Appendix), which therefore controls for potential seasonality and trends in the data. The time series is first smoothed using a 10-day moving average in order to remove the daily

noise and weekly cycle, and better capture the underlying signal. A similar number of days to smooth the time series has been applied in other studies using HFA [5, 10, 15, 22]. We further control for several factors on a daily scale (see S2 Appendix for discussion on the control variables). First, we include the number of confirmed cases as a fraction of the population (Cases), since the severity of the health crisis was found to influence differences in the extent of economic output losses across countries [23]. Second, we include the reduction in demand in trade-dependent countries as a control variable (Demand), as demand reductions, especially when exporting countries are not yet imposing lockdowns, could lower domestic export [6]. Third, we account for the potential reduction in exports due to supply-shortages (Supply) by accounting to what extent changes in imports might affect exports (due to the vertical specialization of domestic supply-chains, see Hummels et al. [27]). For instance, Cerdeiro and Komaromi [22] provide empirical evidence of the transmission of supply disruptions through international maritime trade, which was particular apparent in the early stages of the pandemic. At last, we account for other endogenous factors that are likely to be serially correlated with exports (which we control for by adding a lag of the export change) (Export lag). Alongside the control variables, we add a country fixed effects to account for time-invariant country-specific characteristics that affects exports and a time fixed effects to account for changes in the global economy that drive changes in exports across countries. Reference is made to S2 Appendix for a more detailed description of the econometric model.

From the OxCGRT [24], we obtained information on nine NFI that potentially affect economic activity: C1—School closing; C2—Workplace closing; C3—cancel public events; C4—Restrictions on gatherings; C5—Close public transport; C6—Stay at home; C7—Restrictions on internal movement; C8—International travel controls; H2—Testing policy. We scale the severity of the policies on a scale between 0 and 1, thereby assuming a linear relationship between maritime exports and the severity of policies. Moreover, we create a composite stringency index (Stringency) of all policies (C1-C8) by adding all individual policies together and rescaling the index between 0 and 1.

Given that some non-intuitive choices had to be made for the model specification, we perform a sensitivity analysis of various assumptions made to make our parameter value choices transparent and to test the robustness of the results. We include three sensitivity tests: (1) the number of days used to smooth the time series (10, 7 or 3 days), (2) the time lag of the export change adopted (3, 5 or 7 days), and (3) the inclusion of different time fixed effects (day, week, month). Moreover, we perform two additional robustness checks. The first one is by implementation the lagged effects of NFI on export (no lag, 5 days, 10 days), as it might be that introducing NPI affects export in a lagged manner. Second, we are concerned about the possible multicollinearity between NPI, because the implementation of NPI is often done simultaneously. As an alternative, we introduce the individual NPI one at a time in the model, similar as done in Deb et al. [5], which, despite introducing omitted variables bias, helps to check the robustness of the results. More details can be found in S2 Appendix.

## Results

### Model validation

We find a good fit between the values predicted by our algorithm and the reported trade flows on a port-level (correlation coefficient between 0.52–0.96) and a country-level (correlation coefficient between 0.79–0.98), with a general overestimation for smaller ports, and ports and countries with large trade imbalances (e.g. small islands). For the external validation data, we find correlation coefficients of 0.84–0.86 for the aggregated trade data and 0.73–0.78 for the sector-specific trade data (on a country level). Again, smaller trade flows are harder to predict.

The accuracy of the method is also found to be dependent on the coverage of information in the AIS data (some attributes are manually put in), especially information on the vessel draft, which is less frequently reported in developing countries.

## Port-level trade flows

In the first eight months of 2020, the number of port calls across all ports reduced by 4.4% compared to the same months in 2019. Fig 1A shows the average change in total trade (imports + exports) in terms of volume (in million tonnes, MT) over the months January-August. The vast majority of ports have experienced a decline in total trade, although a number of ports in Brazil, the Gulf of Mexico region, the Middle-East, Australia, and parts of South-Korea and the Philippines have seen an increase in trade in 2020 relative to 2019. The top 20 port with the largest changes in volume in terms of total trade, imports and exports are included in Table 1. The ports with the largest absolute changes in volume are the ports of Ningbo (China, -68.5 MT), Rotterdam (Netherlands, -43.2 MT), Shanghai (China, -32.5 MT), Wuhan (China, -21.6 MT) and Tubarao (Brazil, -20.7 MT). The largest changes in imports are found for the ports of Ningbo (China, -43.5 MT), Rotterdam (Netherlands, -40.1 MT), Shanghai (-22.4 MT), Zhou-shan (China, -22.4 MT) and Amsterdam (Netherlands, -12.2 MT). These ports, and the other ports in the list, function as major gateway ports for a country to import final products (New

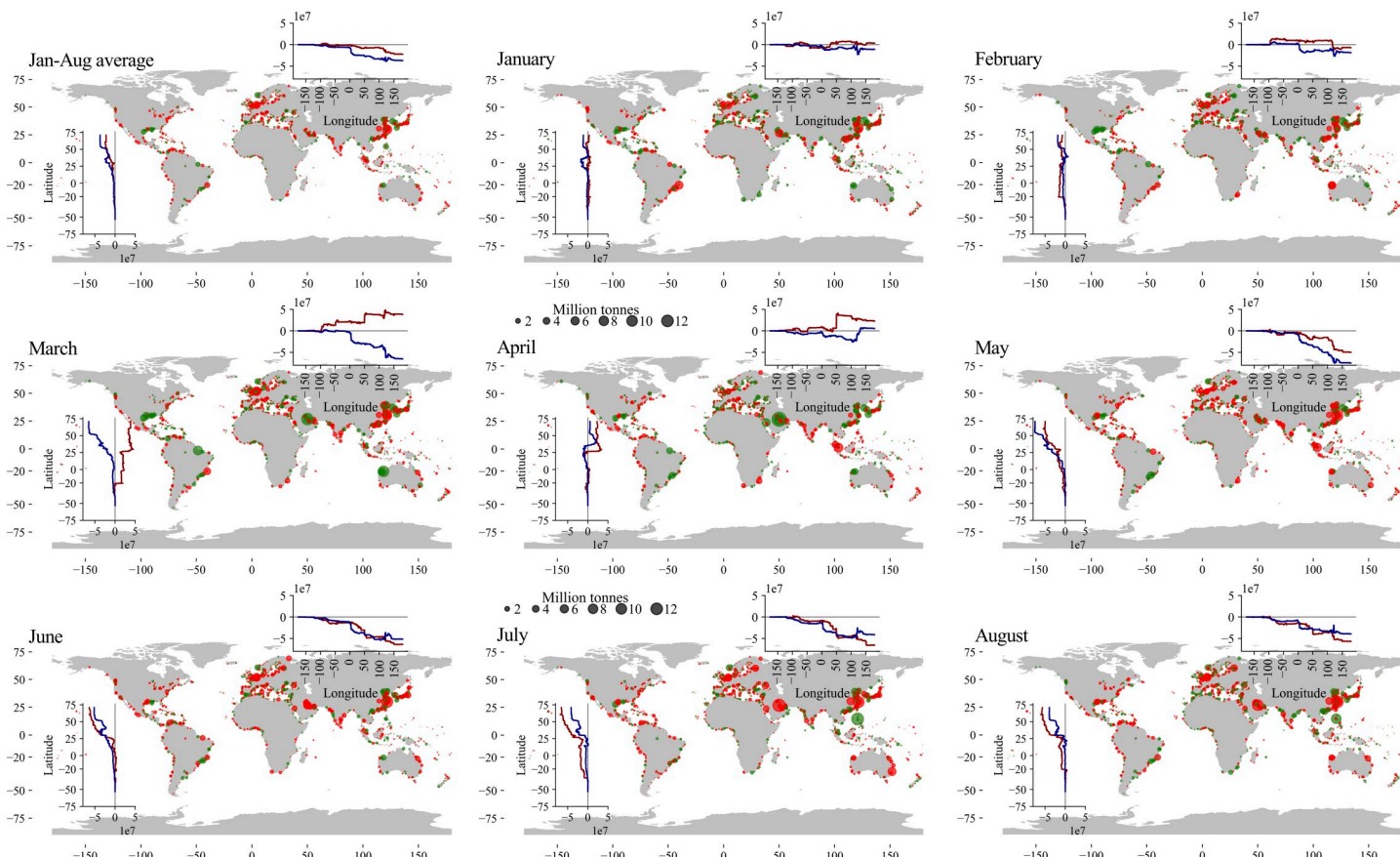

**Fig 1. Port-level trade losses over time.** The geographical location and magnitude of trade losses for Jan-Aug 2020 compared to 2019, including the average over the eight months and the losses per month. Green = positive change, red = negative change. The subplots show the cumulative change over latitude and longitude for imports (dark blue) and exports (dark red). This figure was generated using the 'Geopandas' package (https://geopandas.org) and Python Programming Language (version 3.7). The underlying basemap is derived from 'Natural Earth' global vector data (https://www.naturalearthdata.com).

**Table 1. Largest absolute trade losses on a port-level.**

| | Total trade | | | Imports | | | Exports | | |
|---|---|---|---|---|---|---|---|---|---|
| Rank | Port | iso3 | Change (MT) | Port | iso3 | Change (MT) | Port | iso3 | Change (MT) |
| 1 | Ningbo | CHN | -68.5 | Ningbo | CHN | -43.5 | Ningbo | CHN | -25.0 |
| 2 | Rotterdam | NLD | -43.2 | Rotterdam | NLD | -40.1 | Tubarao | BRA | -17.1 |
| 3 | Shanghai | CHN | -32.5 | Shanghai | CHN | -22.4 | Novorossiysk | RUS | -11.5 |
| 4 | Wuhan | CHN | -21.6 | Zhoushan | CHN | -13.8 | Wuhan | CHN | -10.9 |
| 5 | Tubarao | BRA | -20.7 | Amsterdam | NLD | -12.2 | Beaumont | USA | -10.6 |
| 6 | Zhoushan | CHN | -18.8 | Rizhao | CHN | -11.3 | Dampier | AUS | -10.2 |
| 7 | Amsterdam | NLD | -17.4 | Wuhan | CHN | -10.7 | Shanghai | CHN | -10.1 |
| 8 | Shekou | CHN | -14.2 | Mina Al Ahmadi | KWT | -9.5 | Haypoint | AUS | -9.2 |
| 9 | Hong Kong | HKG | -12.3 | Vlissingen | NLD | -8.5 | Lumut | MYS | -7.6 |
| 10 | Vlissingen | NLD | -12.2 | Zhanjiang | CHN | -7.6 | Shekou | CHN | -7.5 |
| 11 | Singapore | SGP | -12.1 | Umm Said | QAT | -7.4 | Tianjin | CHN | -7.3 |
| 12 | Rizhao | CHN | -11.7 | Yokohama | JPN | -7.3 | Fujairah | ARE | -7.2 |
| 13 | Novorossiysk | RUS | -11.7 | Ghent | BEL | -7.1 | Tangshan | CHN | -6.3 |
| 14 | Lumut | MYS | -11.6 | Singapore | SGP | -6.8 | Xiamen | CHN | -6.1 |
| 15 | Dampier | AUS | -10.8 | Hong Kong | HKG | -6.7 | Itaqui | BRA | -5.8 |
| 16 | Yokohama | JPN | -9.9 | Shekou | CHN | -6.7 | Bohai Bay | CHN | -5.7 |
| 17 | Haypoint | AUS | -9.7 | Krishnapatnam | IND | -6.7 | Puerto Bolivar | COL | -5.7 |
| 18 | Beaumont | USA | -9.5 | Magdalla | IND | -6.6 | Hong Kong | HKG | -5.6 |
| 19 | Ghent | BEL | -9.4 | Port of Le Havre | FRA | -6.4 | Primorsk | RUS | -5.5 |
| 20 | Zhanjiang | CHN | -9.1 | New York-New Jersey | USA | -6.0 | Richards Bay | ZAF | -5.4 |

The top 20 total trade, imports and export losses on a port-level expressed in million tonnes (MT). The losses cover the period Jan-Aug 2020 compared to Jan-Aug 2019.

York-New Jersey, Rotterdam), or are essential for specific supply-chains, such as textiles and electronics manufacturing (Shanghai, Ningbo, Zhoushan), steel and paper manufacturing (Ghent, Amsterdam, Rizhou), car manufacturing (Yokohama) and raw materials (coal imports for Krishnapatnam). The largest export changes are found for the ports of Ningbo (China, -25.0 MT), Tubarao (Brazil, -17.1 MT), Novorossiysk (Russia, -11.5 MT), Wuhan (China, -10.8 MT), Beaumont (USA, -10.6 MT) and Dampier (Australia, -10.2 MT). These ports, and the other top 20 ports with the largest export losses, are all important export ports for global supply-chains, including the exports of iron ore (Dampier, Tubarao), coal (Haypoint), oil and refined petroleum products (Puerto Bolivar, Fujairah, Beaumont and Novorossiysk) and manufacturing products (Ningbo, Wuhan and Shanghai).

Fig 1B–1M show the changes in total trade per month for all ports and the cumulative changes in trade over latitude and longitude. In January, losses are predominantly pronounced in China that extended their Lunar New Year holiday [28], among other measures, resulting in output losses to the Chinese industry. This resulted in a direct demand shock, in particular for the export of raw materials (e.g. iron ore, copper, nickel) that China predominantly imports [29]. This can be observed from the large negative losses found in the large export ports of Brazil. In February, ports in Europe experienced their first drop in imports (blue line top plot), while export losses are still concentrated in Asia. This import drop in Europe coincided with the transit time from China to Europe, which is around three weeks. The export drop is, alongside Brazil, also visible in the main iron ore exporting ports in Australia (Port Hedland and Port Walcott) and South Africa (Port of Richards Bay) that both supply iron ore to the Chinese industry. In March, exports temporally recovered, while imports dropped in many parts of the world, mainly to due to initiation of lockdowns in economies outside Asia. In particular,

India, Malaysia, Singapore, USA West Coast and Mexico saw a large drop in trade in March. In April, trade partly recovered in the Northern Hemisphere, while in May the second drop in global trade hits the global economy, as a widespread reduction in demand and supply ripple through the economy. Losses are again pronounced in China and Western Europe, leading to the lowest total import and exports changes on a global scale. In June, July and August, a partial recovery is visible for some ports, while the Middle-East, Eastern Australia, Japan and Western Europe (in particular Belgium and the Netherlands) show large losses. For the Middle-Eastern countries, the collapse of the oil market has contributed to the large trade losses (which are predominantly exports losses). In August, signs of recovery (especially imports) are visible for the Philippines, India, South Africa, Brazil and Argentina, and parts of the Mediterranean, while other countries are still experiencing large losses.

## Geographical disparity

Fig 2A and 2B show the country-aggregated relative changes in imports and exports, with the top 20 largest negative (relative) changes included in Table 2 and largest negative absolute changes in S1 Table. The top 20 largest total trade losses range between 17–36%. The largest percentage change in imports are associated with small economies such as Turks and Caicos Islands, the Caribbean Netherlands, Bahrain, Anguilla, Federated States of Micronesia and Madagascar (all between 28–37% reduction). Most of the countries with the largest import losses are Small Island Developing States, which are characterised by having large import-dependencies due to their small domestic economies, being reliant on maritime trade flows for trade, and importing large amount of goods to support the tourism sector that constitutes a large share of the country's GDP [30]. With the tourist industry collapsing due to the COVID-19 outbreak [31], the imports are expected to drop significantly, explaining the widespread reductions observed. Other countries, like a number of countries in Africa, Myanmar, Oman, Philippines, the Baltic States and Sweden have increased their imports, likely due to the increased need for food and medical supplies in developing nations or increased household consumption in some developed countries. In terms of exports, the largest relative losses are found for Libya, New Caledonia, Guinea-Bissau, Northern Mariana Islands, Cape Verde and Sudan (all between 50–78% reduction). These countries include many raw materials exporting countries that have suffered from the demand shock across the world, in particular through trade dependencies with Europe, China and the United States [32]. Moreover, many low income countries had to pro-actively lockdown economies to protect their health care system,

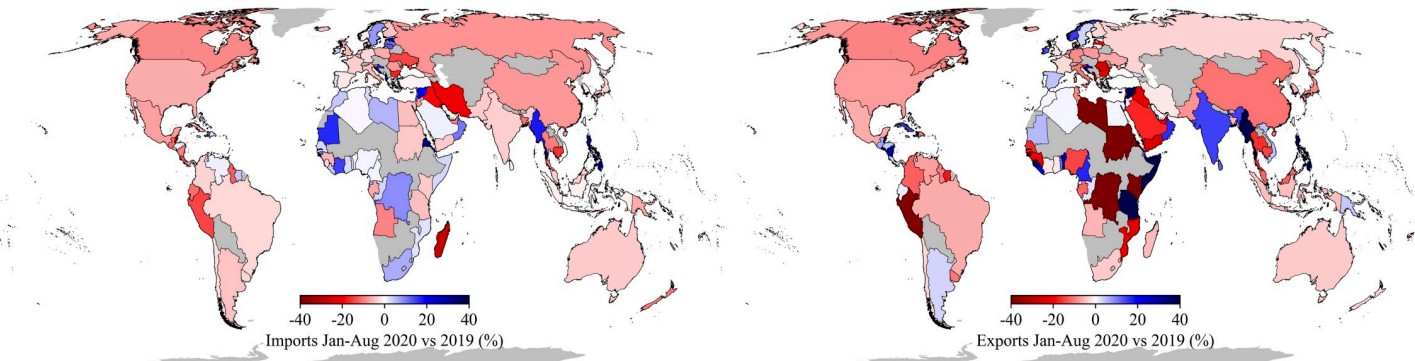

**Fig 2. Country-level relative trade losses.** The relative trade losses for Jan-Aug 2020 compared to 2019 expressed in percentage change. Grey countries indicate no data available. This figure was generated using the 'Geopandas' package (https://geopandas.org) and Python Programming Language (version 3.7). The underlying basemap is derived from 'Natural Earth' global vector data (https://www.naturalearthdata.com).

**Table 2. Largest relative trade losses on a country-level.**

| Rank | Total trade | | Imports | | Exports | |
|------|-------------|----------|---------|----------|---------|----------|
| | Country | Change (%) | Country | Change (%) | Country | Change (%) |
| 1 | Anguilla | -35.6 | Turks and Caicos Islands | -36.9 | Libya | -77.8 |
| 2 | Libya | -34.3 | Bonaire, Saint Eustatius and Saba | -35.9 | New Caledonia | -64.9 |
| 3 | Federated States of Micronesia | -33.5 | Bahrain | -31.3 | Guinea-Bissau | -55.6 |
| 4 | Cape Verde | -30.6 | Anguilla | -30.7 | Northern Mariana Islands | -54.9 |
| 5 | Peru | -28.3 | Federated States of Micronesia | -29.8 | Cape Verde | -53.7 |
| 6 | Bonaire, Saint Eustatius and Saba | -26.8 | Madagascar | -28.4 | Sudan | -49.4 |
| 7 | Malta | -26.2 | Timor-Leste | -26.0 | Montenegro | -45.1 |
| 8 | Eritrea | -26.1 | Malta | -25.7 | Eritrea | -44.6 |
| 9 | Madagascar | -25.1 | Grenada | -22.4 | Dem. Republic Congo | -44.3 |
| 10 | Montenegro | -24.9 | Belize | -22.3 | Vanuatu | -40.0 |
| 11 | Turks and Caicos Islands | -24.8 | Iran | -21.8 | Kenya | -39.6 |
| 12 | Vanuatu | -24.4 | Seychelles | -21.5 | Peru | -39.2 |
| 13 | Seychelles | -23.7 | French Polynesia | -21.1 | Federated States of Micronesia | -38.4 |
| 14 | Timor-Leste | -23.3 | Aruba | -19.8 | American Samoa | -34.9 |
| 15 | Northern Mariana Islands | -22.2 | Vanuatu | -19.4 | Albania | -32.6 |
| 16 | French Polynesia | -21.1 | Iraq | -19.3 | Seychelles | -28.0 |
| 17 | Iraq | -20.1 | Kuwait | -19.0 | Malta | -27.2 |
| 18 | New Caledonia | -19.3 | Macau | -18.3 | Yemen | -25.8 |
| 19 | Bulgaria | -19.1 | Bulgaria | -17.4 | Romania | -25.6 |
| 20 | Romania | -17.6 | Cape Verde | -17.2 | Saint Vincent and the Grenadines | -23.7 |

The total trade, imports and export losses on a country-level expressed in million tonnes (MT). The losses cover the period Jan-Aug 2020 compared to Jan-Aug 2019.

or are engaged in economic activities that are less able to be done remotely [32, 33]. Some countries have increased their exports, such as India, Myanmar, Vietnam and Philippines, potentially because of production shifts of manufacturing goods to these countries when factory shut down in China [34]. Moreover, exports grew in Argentina, mainly due to booming exports of food products (e.g. soybeans, beef) to the United States and China [35], and in Tanzania, which increased its exports of gold and food (e.g. nuts) and textile products (e.g. cotton) [36].

Using the World Bank income classification (2019–2020), we test whether high and upper middle income countries have experienced more severe impacts than low and lower middle income countries. Without excluding outliers from the data, we find a significant difference (two-sided t-test with $p > 0.05$) between both income groups for exports and imports, with high and upper middle income countries having higher export losses. Hence, although the high and upper middle income countries have higher mean export losses, the most extreme export losses and gains are found for low and lower middle income countries.

## Time series of total and sector-specific trade changes

The total trade losses are not uniform across sectors. Fig 3 shows the estimated total trade losses over time (Fig 3A) together with the trade losses for the 11-sector classification considered. The total trade losses are found to be between -7.0% and -9.6% (mean -8.3%), which is equal to around 206–286 MT in volume losses and up to 225–412 billion USD in trade value (uncertainty due to differences in total import and export losses and due to the volume to

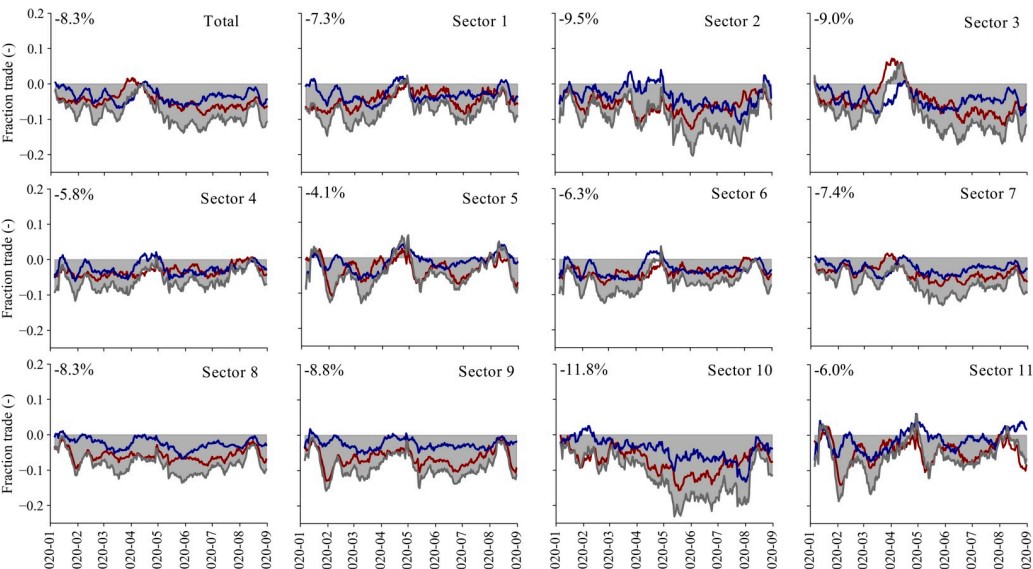

**Fig 3. Sector-specific losses over time.** The change in daily global total trade as a fraction of the average daily trade (over 2019). The dark blue line represent imports, the dark red line represent exports, whereas the grey line indicate total trade (import + exports). Sector 1: Agriculture; Sector 2: Fishing; Sector 3: Mining and quarrying; Sector 4: Food and beverages; Sector 5: Textiles and wearing apparel; Sector 6: Wood and paper; Sector 7: Petroleum, chemical and non-metallic mineral products; Sector 8: Metal products; Sector 9: Electrical and machinery; Sector 10: Transport equipment; Sector 11: Other manufacturing.

value conversion, see S1 Appendix). The time series show (Fig 3A) a clear initial drop in trade in the first three months, after which trade partly recovers, followed by a second, more pronounced, drop in trade. In late August 2020, a sign of economic recovery is not yet visible.

Some supply-chains have been more resilient than others. The most resilient sectors are found to be Textiles and wearing apparel (-4.1%), Food and beverages (-5.8%), Other manufacturing (-6.0%) and Wood and paper (-6.3%). The times series of Textiles and wearing apparel and Other manufacturing show, however, a large drop in exports in the early stages of the pandemic, mainly associated with production in China and other Asian economies (e.g. Bangladesh, Malaysia), followed by a gradual recovery and a less steep second drop. The Wood and paper and Food and beverages sectors have been more stable throughout pandemic out-break, as supply-chains were not significantly disrupted, and demand for products only gradually declined, followed by signs of a recovery at the end of August. The largest relative changes are found for the Fishing sector (-9.5%), Mining and quarrying (-9.0%), Manufacturing of electronics and machinery (-8.8%), and Manufacturing of transport equipment (-11.8%). The drop in fishing products peaked late in the pandemic with a clear recovery in July and August. The time series of the mining and quarrying sector shows a more complex picture with a sharp drop in the beginning of the pandemic, as demand for raw materials decreased in Asia, fol-lowed by a steep increase in trade to restock inventories, after which a total collapse of the mar-ket can be observed, mainly associated with reduced demand for oil. For the two manufacturing sectors, the large losses are the result of significant supply-chain disruptions that caused upstream production processes to halt due to a shortage in supplies [37]. In partic-ular the Transport manufacturing industry, characterised by just-in-time logistic services and highly specialised production processes, experienced a gradual disruption throughout the first few months, after which trade declined more than 20% in May, June and July.

**Table 3. The results of the various regression models.**

| Parameter | Model1 Beta | Model2 Beta | Model3 Beta | Model4 Beta |
|---|---|---|---|---|
| Composite | -4.003** | | -4.726** | |
| C1 | | -3.968*** | | -7.212*** |
| C2 | | -2.891** | | -0.884 |
| C3 | | 0.012 | | 0.547 |
| C4 | | -2.375* | | -3.431** |
| C5 | | -3.864*** | | -5.857*** |
| C6 | | 3.857** | | 7.781*** |
| C7 | | 2.729** | | 1.896 |
| C8 | | 3.347*** | | 2.602 |
| H2 | | 0.960 | | -0.166 |
| Demand | -0.021 | -0.026 | 0.021 | 0.013 |
| Cases | -0.167** | -0.179*** | -0.658*** | -0.682** |
| Supply | 0.068*** | 0.068*** | 0.061*** | 0.062*** |
| Export lag | 0.373*** | 0.372*** | 0.363*** | 0.360*** |
| R2 | 0.348 | 0.350 | 0.354 | 0.357 |
| R2-adjusted | 0.337 | 0.339 | 0.339 | 0.342 |
| F-statistic | 32.41 | 31.77 | 24.46 | 24.10 |

The table shows the estimated beta coefficients and goodness of fit statistics for the six model specifications discussed.

*$p < 0.1$

**$p < 0.05$

***$p < 0.01$.

## Impact of non-pharmaceutical interventions on export

The results of the panel regression model are included in Table 3. As described in the methodology, the base model (Model 1 and 2) includes country and time (day) fixed effects and daily control variables for the number of confirmed cases as a fraction of the population (Cases), demand reduction in trade dependent countries (Demand), the potential supply disruptions through changes in import that are used for exports (Supply), and potential other factors that are autocorrelated with the change in exports (Export lag).

The effect of the composite index on daily export change is strong, and statistically significant ($p < 0.01$), with a 10% increase of the index resulting in a -0.40% change in exports (Model 1). The influence of NPI on exports is mixed with some measures showing a negative impact while others showing a positive impact (Model 2). Negative impacts are found for school closing (C1, -3.97%, significant at $p < 0.01$), workplace closure (C2, -2.89%, significant at $p < 0.05$), restrictions on public gatherings (C4, -2.38%, significant at $p < 0.10$), and closing of public transport (C5, -3.86% significant at $p < 0.01$). Surprisingly, a positive effect is found for stay at home requirements (C6, +3.86%, significant at $p < 0.05$), restrictions on internal movement (C7, +2.73%, significant at $p < 0.05$) and restrictions on international travel (C8, +3.45%, significant at $p<0.01$). Additionally, we run a model (Model 3 and 4) which include only the days where the outbreak become significant in a country (which we define as having at least 50 confirmed cases). The coefficient of the composite index is found to be slightly higher (-4.73% relative to -4.00%, both significant at $p < 0.05$). For the individual NPI, the effect of school closures, restrictions on public gatherings, closures of public transport and stay at home policies becomes larger, while the effect of workplace closures, restrictions on internal

movement and restrictions on international travel diminishes becomes not significant. This difference suggest that some policies particularly affected the economy when implemented pro-actively (before the health crisis started), while it becomes less important when cases become more prevalent in a country. For instance, 52 countries implemented pro-active work-place closures before reaching 50 positive cases.

Next, we test the robustness of the results by changing a number of assumptions in the model (see S2 Appendix for full details). First, we test whether the 10-day smoothing of the time series affects the results by evaluating the results for a 7-day and 3-day smoothing period. The results are shown in S2 Table 1 in S2 Appendix, and shows that the effect of the composite index becomes less strong for a 7 day period and becomes not significant for a 3 day period (although with similar values). For the individual NPI, the results for the closures of schools and public transport, and restrictions on internal movement and international travel are robust across models, while the other NPI become not significant. Hence, less smoothing of the time series lowers the signal to noise ratio, making it generally harder to detect the influence of NPI. Second, we vary the lag of the export change (3 day, 5 day and 7 day), with the results included in S2 Table 2 in S2 Appendix. Across all models, increasing the lag will increase the effect of the composite index and the individual NPI, since the influence of other endogenous factors becomes less strong (and more weight is attributed to the NPI). Therefore, the results presented in Table 3 are considered the most conservative values. Third, we test whether the inclusion of alternative time fixed effects (day, week and month) in the model changes the result (see S2 Table 3 in S2 Appendix). The results do not change much, although the composite index becomes larger and more significant.

For the first robustness checks, we test whether implementing the policies in a lagger manner changes the result. We lag the policies 5 and 10 days and evaluate whether the estimated coefficients become larger, which would indicate that the NPI indeed influence the export dynamics in a lagged way (S2 Table 4 in S2 Appendix). For school closures, stay at home requirements and restrictions on international travel, the effect becomes larger, suggesting that the influence of these NPI takes days to materialize. For the second robustness check, we contrast the coefficient of the NPI when included altogether or one at a time (S2 Table 5 in S2 Appendix). The negative coefficients of C1, C2, C4 and C5 are still found when including only these policies (and significant at $p < 0.01$), whereas the positive effects of C6 and C7 become small and not significant. The positive effect of C8 is found at $p < 0.1$. Thus, the interpretation of the positive effects of C6-C8 should be done with caution.

## Discussion and conclusion

We present a near-global analysis of maritime trade indicators based on empirical vessel track-ing data, which we use as a high-frequency indicator of economic activity. We illustrate how the implementation of NPI have resulted in large trade losses with a strong geographical and sectoral heterogeneity, with individual NPI affecting the economy in different ways.

Our estimate of a 4.4% reduction in global ports calls for the first eight months of 2020 is lower than the 8.7% predicted by UNCTAD for the first six months [38]. The main reason for this difference is associated with the inclusion of different vessel types. Whereas we include only the main trade-carrying vessels, the UNCTAD analysis also included passengers vessels (66% of total port calls), which have seen the largest drop in port calls (-17% for passenger ves-sels). Moreover, the sector-level trends we found are in line with the sector-level impacts (based observed trade data of China, the European Union and the United States) for the first quarter (Q1) of 2020 as presented in the UNCTAD analysis [38], that stated that in particular the automotive industry (-8%), machinery (-8%), office machinery (-8%) and textiles and

apparel (-11%) are particularly hit. Our analysis, which differs by only including maritime trade (instead of total trade) and having a global scale (instead of three regions), found the average losses in Q1 for the textiles and apparel (Sector 5), electrical equipment and machinery manufacturing (Sector 9), transport equipment (Sector 10) and other manufacturing (Sector 11) to be respectively 6.2%, 9.2%, 6.5% and 8.4%. The trade losses we estimate (225–412 billion USD for the first eight months) are considerably lower than reported in the modelling framework of Lenzen et al. [39], who estimated the trade losses for the first five months of 2020 to be 536 billion USD. Again, part of this difference is due to the coverage of countries (they provide a full global analysis) and modes of transport (all modes compared to maritime only). Still, input-output based analysis, as done in Lenzen et al. [39], often fail to consider adaptative behaviour in the global economic system, which can dampen economic impacts [40]. Our result of a contraction of almost 10% of maritime trade is also in line with the most recent UNCTAD trade report [41] that showed that during the first three quarters of 2020, global trade (in value terms) decreased by 8%, with the largest hit to trade in Q2 of 2020.

The results of the econometric model provide an alternative view to previous studies that have evaluated the effect of NPI on the spread of the virus [1, 2, 4, 42] and the economy [5, 8]. We find clear evidence of the negative impacts of NPI on changes in daily exports, with a 10% increase in the overall stringency value resulting in an approximate 0.40% decrease in daily maritime exports. In particular, we find evidence that closures of school, workplaces and public transport had a negative effect on daily exports, whereas stay at home policies, restrictions on internal movements and international travel controls have resulted in a positive effect on daily exports. The stark negative effect of school closures, robust across model specifications, is in line with previous model-based estimates that showed that school closures, resulting in a large forced absenteeism by working parents, have large impacts on the economy. For instance, a study [43] showed that school closures of 12–13 weeks in the United Kingdom would result in a 0.2–1% drop in national GDP, whereas another study [44] showed that a 8-week school closure in the United States could result in losses of up to 3% of GDP. Looking at the relative effect of different NPI, our results are in partial agreement with Deb et al. [5], who used nitrogen emissions as an indicator of industrial output, and found that workplace closures and closures of public transport had the largest influence on the drop in emissions, followed by cancellation of events and school closures. However, direct comparison with these results should be taken with caution, as the causal mechanisms may differ. For instance, changes in nitrogen emissions primarily reflect emissions from heavy industry and transportation, while changes in maritime exports reflect industrial activity across all sectors. The large drop in emissions from the cancellation of events, for instance, is most likely associated reduced emissions from transport to and from events, but might not necessarily affect industrial exports much (as we find no evidence that this influenced exports). Similarly, school closures, stay at home requirements and restrictions on internal movements had the highest effect on changes in mobility across Europe, as shown in Santamaria et al. [15], which can also be attributed to different causal pathways. We find that pro-active business closures (before cases were high in a country), had a negative effect on exports, but its effect becomes insignificant when case rates are high in a country. This could be related to the fact that many countries (in particular countries with below-average GDP/capita) had to pro-actively close workplace to protect their health system. More specifically, 75% of low-income countries and 62% of lower-middle income countries in our sample had to implement such pro-active workplace closures, compared to 30% of high-income countries. This is in agreement with Furceri et al. [23] who found that countries with lower GDP/capita had higher output losses relative to the extent of the health crises. The possible causal pathways resulting in positive effects (compared to countries not imposing these NPI) of stay at home policies, restrictions on internal

movements and bans on international travel are difficult to establish. One reason could be that these policies helped containing the spread of the virus, while causing minimal disruptions to economic activity, resulting in relative positive effects. Fezzi and Fanghella [9] also concluded that pursuing a herd immunity strategy did not shield countries from economic impacts, in line with our results. Moreover, Furceri et al. [23] find a link between the deaths per capita and reduced industrial output, showing how implementing targeted policies to reduce the transmission might protect the economy. Still, the positive effects should be further scrutinized, as was shown in the second robustness check, before we can draw a definite conclusion on this.

Overall, all results should be interpreted with caution, as many factors could potentially influence these causal relationships. For instance, temporal increases in maritime transport during some periods of the pandemic could be driven by the large increase in trade of medical supplies (e.g. PPE) and mode substitution from air to maritime [45], irrespective if policies were imposed during these periods. Therefore, testing alternative economic indicators, such as data on mobility, energy consumption and nitrogen emissions, as done in Deb et al. [5], can help support these findings. Future work can refine our estimates by adding more economic data when it becomes available, including extending the analysis to indicators of air, road and rail transport. Moreover, the empirical estimates derived here can be used to constrain and validate macro-economic impact models, as used in previous work [6, 39], in order to improve the quantification of the total losses to industrial output as the pandemic unfolds.

In short, our analysis of the economic implications of introducing NPI into society can help evaluate the cost-benefit of the different NPI, which may help governments construct effective portfolios of policies as many countries enter a second or third wave of COVID-19 cases [46]. Moreover, we emphasize how real-time indicators of economic activity, such as maritime trade, can help monitor the unfolding economic disruptions across spatial scales and support governments and international organisations in their economic recovery efforts by allocating funds to the hardest hit economies and sectors.

## Supporting information

**S1 Appendix. Methodology maritime trade estimates.**
(PDF)

**S2 Appendix. Econometric model.**
(PDF)

**S1 Table. The top 20 largest negative maritime trade losses on a country-level.** The total trade, imports and exports losses expressed in million tonnes (MT). The losses cover the period Jan-Aug 2020 compared to Jan-Aug 2019.
(PDF)

## Acknowledgments

The authors would like to thank the United Nations Statistical Division and the UN Global Working Group on Big Data for Official Statistics, in particular Markie Muryawan and Ronald Jansen, for providing the AIS data.

## Author Contributions

**Conceptualization:** Jasper Verschuur, Elco E. Koks, Jim W. Hall.

**Formal analysis:** Jasper Verschuur.

**Funding acquisition:** Jim W. Hall.

**Methodology:** Jasper Verschuur, Elco E. Koks, Jim W. Hall.

**Supervision:** Elco E. Koks, Jim W. Hall.

**Writing – original draft:** Jasper Verschuur.

**Writing – review & editing:** Elco E. Koks, Jim W. Hall.

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
