## [Decision Letter · Decision Letter 0]

29 Jan 2021

PONE-D-20-33893

The implications of large-scale containments policies on global maritime trade during the COVID-19 pandemic

PLOS ONE

Dear Dr. Verschuur,

Thank you for submitting your manuscript to PLOS ONE. After careful consideration, we feel that it has merit but does not fully meet PLOS ONE’s publication criteria as it currently stands. Therefore, we invite you to submit a revised version of the manuscript that addresses the points raised during the review process.

We look forward to receiving your revised manuscript.

Kind regards,

Bing Xue, Ph.D.

Academic Editor

PLOS ONE

Journal Requirements:

2. Would other researchers in principle be able to partner with the UN Global Platform AIS Task Team to obtain the same or a similar dataset? If so, please include this information in your Data availability statement, including any contact information. Please also include in your Data availability statement references to the other data sources used.

3. We noted in your submission details that a portion of your manuscript may have been presented or published elsewhere.

"Yes, we have written a Matters Arising on a paper published in Nature Human Behaviour (Guan et al., 2020) with a small subset of the data. However, no formal analysis was performed for this publication, only a commentary on the aforementioned article. Therefore, we do not see any reason how this might influence this publication "

Please clarify whether this publication was peer-reviewed and formally published. If this work was previously peer-reviewed and published, in the cover letter please provide the reason that this work does not constitute dual publication and should be included in the current manuscript.

4. We note that Figures 1-3 in your submission contain map images which may be copyrighted. All PLOS content is published under the Creative Commons Attribution License (CC BY 4.0), which means that the manuscript, images, and Supporting Information files will be freely available online, and any third party is permitted to access, download, copy, distribute, and use these materials in any way, even commercially, with proper attribution. For these reasons, we cannot publish previously copyrighted maps or satellite images created using proprietary data, such as Google software (Google Maps, Street View, and Earth). For more information, see our copyright guidelines: http://journals.plos.org/plosone/s/licenses-and-copyright.

4.1.    You may seek permission from the original copyright holder of Figures 1-3 to publish the content specifically under the CC BY 4.0 license. 

4.2.    If you are unable to obtain permission from the original copyright holder to publish these figures under the CC BY 4.0 license or if the copyright holder’s requirements are incompatible with the CC BY 4.0 license, please either i) remove the figure or ii) supply a replacement figure that complies with the CC BY 4.0 license. Please check copyright information on all replacement figures and update the figure caption with source information. If applicable, please specify in the figure caption text when a figure is similar but not identical to the original image and is therefore for illustrative purposes only.

5. We note you have included a table to which you do not refer in the text of your manuscript. Please ensure that you refer to Table 3 in your text; if accepted, production will need this reference to link the reader to the Table.

Reviewers' comments:

Reviewer's Responses to Questions

**Comments to the Author**

1. Is the manuscript technically sound, and do the data support the conclusions?

Reviewer #1: Yes

Reviewer #2: Partly

2. Has the statistical analysis been performed appropriately and rigorously? 

Reviewer #1: Yes

Reviewer #2: Yes

3. Have the authors made all data underlying the findings in their manuscript fully available?

Reviewer #1: Yes

Reviewer #2: Yes

4. Is the manuscript presented in an intelligible fashion and written in standard English?

Reviewer #1: Yes

Reviewer #2: Yes

5. Review Comments to the Author

Reviewer #1: Dear Author,

The statistical analysis of your manuscript has been performed appropriately and rigorously. However, an addition needs to be made in this regard. The explanation as to why the analysis method was chosen should be persuasive to the reader. The explanation in the line between 105-108 does not provide an adequate explanation for why the fixed effects model was used. And this explanation is not seen elsewhere in the study. this point should be explained.

Other matters written in the “discussion and conclusion” part were appropriate and explanatory, but clear emphasis should be placed on the regression model results. Those who are interested in the application (panel) part of the study will want to see the inference of the authors and be informed in the conclusion part. For example “an inference can be made based on the main finding of the study (panel) or a discussion can be conducted on why this fundamental finding is so.”

Apart from this, the topics you need to review / review about the article are listed below :

- line 363 : Table 1. Please check. Should it be Table 3? (note : And another results given in “S2 Table 1")

- line 363 : C1, -2,63% Please check the number. Some numbers in the same sentence belong to Model 2 and some to Model 4. It should be checked.

- line 376 : C2, -4,76% Please check the number. Some numbers in the same sentence belong to Model 2 and some to Model 4. It should be checked.

- line 378 : C6, +2,74% Please check the number. Some numbers in the same sentence belong to Model 2 and some to Model 4. It should be checked.

Reviewer #2: Reviewer: This paper investigates The implications of large-scale containments policies on global maritime trade during the COVID-19 pandemic. The structure of the article is not very clear and the regression model needs to be completed. My comments are as follows:\\\\

1) Title: The title is too general and is not attractive. The economic impacts is in the keywords but cannot be informed in the title.

2) Keywords：the keywords need to be more concise and specified, the algorithm developed in this paper should be added.

3) Introduction: The introduction is organized with order, but to give readers a better picture I suggest the author to re-structure this section as following order: (i) Your research idea (aim/hypothesis); (ii) Why is it important? (iii) What is new about your work?; (iv) Your approach; (v) Findings & contributions

4) What is the contribution of this paper? The innovation points of the research are not clear.

5) Method: In S2 Appendix: econometric model, Before adopting the fixed effect panel regression model in this paper, is it necessary to perform test(e.g. Hausmann test) on panel data to determine whether to use a fixed effects model, a random effects model or a mixed regression model?

6) In S2 Appendix: econometric model. How to choose the control variables? What are the economic implications or relevant references?

7) The data “67 ports” should be in “the United States” in page 2 line 2, S1_appendix 1.

8) How to choose the data smoothing methods, a 10 days moving average？Does it have specific meaning in the economic field?

9) In S2 Appendix: econometric model. How to choose the value of day lag? (a three day lag of the export change Δ,−Δ)

6. PLOS authors have the option to publish the peer review history of their article (what does this mean?). If published, this will include your full peer review and any attached files.

Reviewer #1: No

Reviewer #2: No

---

## [Author Response · Author response to Decision Letter 0]

19 Feb 2021

Reviewer #1: Dear Author,

-The statistical analysis of your manuscript has been performed appropriately and rigorously. However, an addition needs to be made in this regard. The explanation as to why the analysis method was chosen should be persuasive to the reader. The explanation in the line between 105-108 does not provide an adequate explanation for why the fixed effects model was used. And this explanation is not seen elsewhere in the study. this point should be explained.

-Other matters written in the “discussion and conclusion” part were appropriate and explanatory, but clear emphasis should be placed on the regression model results. Those who are interested in the application (panel) part of the study will want to see the inference of the authors and be informed in the conclusion part. For example “an inference can be made based on the main finding of the study (panel) or a discussion can be conducted on why this fundamental finding is so.”

We thank the reviewer for these comments. To start, we have now changed the how and why of the conclusion/discussion in line with both reviewers’ comments. We agree that there was not enough emphasis on this. Also, at the time of writing, there was limited evidence available to explain the ‘why’ with high confidence (merely anecdotal evidence), but given an influx of new research, this can now be done in a better way. We have therefore changed the Conclusion/Discussion in accordance with this. Moreover, we have changed the Introduction/Problem Statement to better reflection how our research compares with other work what the research gap we are trying to fill. 

On the use of the econometric model, we admit that by writing the methodology in a concise manner we did not expand enough on the justification of the model itself and the control variables used. Moreover, as remarked by reviewer two, some modelling assumptions are not clearly explained or tested. Therefore, to improve the clarity and transparency of the model set-up throughout the manuscript, we have added the following aspects:

We added a short justification for adopting the fixed-effect model in the main manuscript (L170-172) and a more extensive description in the Appendix S2. In short, to decide what model to use, in particular the choice between a random effects and fixed effects model, we performed a Hausman test to test whether we could use a random effects model or had to use a fixed effects model. For both the model with the composite index and with the individual policies we found that the null hypothesis was rejected at p = 0.05 (and at p = 0.1), implying that omitted country-specific variables are correlated with the explanatory variables. Therefore, we are forced to use a fixed-effects model to prevent bias in our model fit. We have provided some explanation what the omitted variables may be. 

We have added a justification per control variable in the main text (see L178-195) and a more extensive description in Appendix S2. 

We have clearly indicated the three non-intuitive model assumptions we made: (1) the number of days of smoothing applied to the time series, (2) the time lag used for the export change in the regression model, and (3) the use and implementation of a time fixed effects. For these three assumptions, we now included a sensitivity analysis to show how changing these assumptions influences the results. 

We have included two types of robustness checks; (1) the potential lagged effect of the implementation of policies on exports and (2) the issue of multicollinearity in the policies when including them altogether in the regression model. 

Moreover, we re-evaluated the base model we adopted in Table 3. First, we initially included a 7 day lag for the effect of the ‘Supply’ control variable. However, because this adds an additional non-intuitive modelling decision to the model set-up, and because there is no clear guidance what value to adopt for this, we decided to remove this lag. This has only a small, and negligible, impact on the results. Second, the initial base model did not include any time fixed effects. However, we believe that this is incorrect, as it became quite clear that the uncertainty in the global economy has influenced the behaviour and confidence of companies and markets. Therefore, our base model now includes a day fixed effects, and we evaluate whether including a week or month fixed effects changes the results (see sensitivity analysis).

More specific responses to your points raised are answered below. 

Apart from this, the topics you need to review / review about the article are listed below :

- line 363 : Table 1. Please check. Should it be Table 3? (note : And another results given in “S2 Table 1")

This was indeed wrongly referenced. We have corrected this now. 

- line 363 : C1, -2,63% Please check the number. Some numbers in the same sentence belong to Model 2 and some to Model 4. It should be checked.

Thank you for pointing this out, we have now made this distinction clear. 

- line 376 : C2, -4,76% Please check the number. Some numbers in the same sentence belong to Model 2 and some to Model 4. It should be checked.

Thank you for pointing this out, we have now made this distinction clear. 

- line 378 : C6, +2,74% Please check the number. Some numbers in the same sentence belong to Model 2 and some to Model 4. It should be checked.

Thank you for pointing this out, we have now made this distinction clear. 

-Reviewer #2: Reviewer: This paper investigates The implications of large-scale containments policies on global maritime trade during the COVID-19 pandemic. The structure of the article is not very clear and the regression model needs to be completed. My comments are as follows:\\\\

We thank the reviewer for the constructive feedback and points to improve the clarity and interpretation of the results. We agree that the introduction was not well structured and the problem statement was missing. We have now rewritten the introduction to make the importance of the work and our contribution relative to other work clear. We hope that this has improved the readability of the paper. 

On the use of the econometric model, we admit that by writing the methodology in a concise manner we did not expand enough on the justification of the model itself and the control variables. Moreover, as remarked by the reviewer, some modelling assumptions are not clearly explained or tested. Therefore, to improve the clarity and transparency of the model set-up throughout the manuscript, we have added the following aspects:

-We added a short justification for adopting the fixed-effect model in the main manuscript (L170-172) and a more extensive description in the Appendix S2. See our comment below for the specifics on this. 

-We have clearly indicated the three non-intuitive model assumptions we made: (1) the number of days of smoothing applied to the time series, (2) the time lag used for the export change in the regression model, and (3) the use and implementation of a time fixed effects. For these three assumptions, we now included a sensitivity analysis to show how changing these assumptions influences the results. 

-We have included two types of robustness checks; (1) the potential lagged effect of the implementation of policies on exports and (2) the potential issue of multicollinearity in the policies when including them altogether in the regression model. 

Moreover, we re-evaluated the base model we adopted in Table 3. First, we initially included a 7 day lag for the effect of the ‘Supply’ control variable. However, because this adds an additional non-intuitive modelling decision to the model set-up, and because there is no clear guidance what value to adopt for this, we decided to remove this lag. This has only a small impact on the results. Second, the initial base model did not include any time fixed effects. However, we believe that this is incorrect, as it became quite clear that the uncertainty in the global economy has influenced the behaviour and confidence of companies and markets. Therefore, our base model now includes a day fixed effects, and we evaluate whether including a week or month fixed effects changes the results (see sensitivity analysis). 

More specific responses to your points raised are answered below. 

-1) Title: The title is too general and is not attractive. The economic impacts is in the keywords but cannot be informed in the title.

Thank you for this comment, we have now changed the title to:

“Global economic impacts of COVID-19 lockdown measures stand out in high-frequency shipping data”

-2) Keywords：the keywords need to be more concise and specified, the algorithm developed in this paper should be added.

We have changed the key words to:

COVID-19; economic impacts; trade prediction algorithm; high-frequency data; non-pharmaceutical interventions

However, key words will not be published online and are only used internally by PlosOne. 

-3) Introduction: The introduction is organized with order, but to give readers a better picture I suggest the author to re-structure this section as following order: (i) Your research idea (aim/hypothesis); (ii) Why is it important? (iii) What is new about your work?; (iv) Your approach; (v) Findings & contributions

We agree that the introduction was messy and the problem statement, in particular our contribution, was not clear. We have rewritten the introduction, also better reflecting on recently published Working Papers and peer-reviewed articles, which has helped us to better position our research with respect to other published or forthcoming research.

-4) What is the contribution of this paper? The innovation points of the research are not clear.

As described above, we have revised the problem statement to better reflect the innovative part of it.

-5) Method: In S2 Appendix: econometric model, Before adopting the fixed effect panel regression model in this paper, is it necessary to perform test(e.g. Hausmann test) on panel data to determine whether to use a fixed effects model, a random effects model or a mixed regression model?

We indeed performed a Hausman test to test whether we could use a random effects model or had to use a fixed effects model (we thought this was common knowledge, but should have been mentioned). For both the model with the composite index and with the individual policies we found that the null hypothesis was rejected at p = 0.05 (and at p = 0.1), implying that omitted country-specific variables are correlated with the explanatory variables. Therefore, we are forced to use a fixed-effects model to prevent bias in our model fit. 

We have now included this statement in the main manuscript and a more detailed description in S2 Appendix to make this clear for the reader.

-6) In S2 Appendix: econometric model. How to choose the control variables? What are the economic implications or relevant references?

The control variables are chosen for two reasons: (1) economic grounds, (2) available data on a global scale. Concerning (1), we apologise for not making the decisions we have made clear enough. At the time of writing the first draft, the evidence to support the adoption of some of the control variables was often merely anecdotal. However, some recent work provides support for the control variables included in our model and this has now been included in both the Method section (see Lines 178-193) and Appendix S2. Concerning (2), we believe there might be other control variables that could help improve the results, but we could not find better suited variables that are available on a global scale and with a daily time step.

-7) The data “67 ports” should be in “the United States” in page 2 line 2, S1_appendix 1.

This was a typo, we have changed this now. 

-8) How to choose the data smoothing methods, a 10 days moving average？Does it have specific meaning in the economic field?

-To the best of our knowledge, there is are no rules or best practises for choosing the smoothing method nor the number of days for the moving average. Applying a moving average (m.a.) has also been adopted in other studies that use high-frequency data to evaluate the effect of lockdown measures, and are often 5-14 days (see citations in Lines 177-178). 

-Compared to, for instance, weekly averages to smooth a daily time series, m.a. has the benefit of better taking into consideration when policies were implemented (on day 1 or day 7 of the week) and when a supply-shock arrived at a country’s ports. The reason for adopting the 10 day m.a. is twofold: after testing different options, we find that a 10 day m.a. effectively filters out the noise for most countries. Second, a 10 day m.a. allows us to filter out a weekly cycle that is present in some countries (e.g. slightly more or less trade during certain days, such as weekend days). 

We have added the follow sentence to the manuscript L175-178 and more details in Appendix S2:

“The time series is first smoothed using a 10-day moving average in order to remove the daily noise and weekly cycle, and better capture the underlying signal. A similar number of days to smooth the time series has been applied in other studies using HFA [5,10,15,22].”

-We do point out that adding a m.a introduces autocorrelation in our error term, but because the fixed-effect model clusters standard errors at the country-level, this will not influence our model fit (as this makes it robust against autocorrelation). We have added this point to Appendix S2. 

- As mentioned above, in order to assess to what extent this assumption influences the results we run some additional models that have a 7 day and 3 day time lag. In short, for a 7 day time lag most effects become slightly weaker, but still hold, whereas for a 3 day lag the noise becomes large, making it harder to detect the effect of the individual policies. 

-9) In S2 Appendix: econometric model. How to choose the value of day lag? (a three day lag of the export change Δ,−Δ)

Similar as above, we agree that this decision is not intuitive and not transparent. We have therefore decided to also include this as a sensitivity test where we test different lags. In general, increasing the lag will lower the effect of this term on exports (as it becomes less serially correlated) and hence will increase the coefficients of the NPI. Since we do not know the ‘correct’ value for this lag, we report only the most conservative values in the main text and add a table with the alternative models in Appendix S2.

---

## [Decision Letter · Decision Letter 1]

8 Mar 2021

Global economic impacts of COVID-19 lockdown measures stand out in high-frequency shipping data

PONE-D-20-33893R1

Dear Dr. Verschuur,

We’re pleased to inform you that your manuscript has been judged scientifically suitable for publication and will be formally accepted for publication once it meets all outstanding technical requirements.

Kind regards,

Bing Xue, Ph.D.

Academic Editor

PLOS ONE

Additional Editor Comments (optional):

Reviewers' comments:

Reviewer's Responses to Questions

**Comments to the Author**

1. If the authors have adequately addressed your comments raised in a previous round of review and you feel that this manuscript is now acceptable for publication, you may indicate that here to bypass the “Comments to the Author” section, enter your conflict of interest statement in the “Confidential to Editor” section, and submit your "Accept" recommendation.

Reviewer #1: All comments have been addressed

Reviewer #2: All comments have been addressed

2. Is the manuscript technically sound, and do the data support the conclusions?

Reviewer #1: Yes

Reviewer #2: Yes

3. Has the statistical analysis been performed appropriately and rigorously? 

Reviewer #1: Yes

Reviewer #2: Yes

4. Have the authors made all data underlying the findings in their manuscript fully available?

Reviewer #1: Yes

Reviewer #2: Yes

5. Is the manuscript presented in an intelligible fashion and written in standard English?

Reviewer #1: Yes

Reviewer #2: Yes

6. Review Comments to the Author

Reviewer #1: All comments have been adressed, thanks you. Authors provided the necessary details regarding the analysis method. The warnings in the conclusion part have been fulfilled. In addition, the errors stated in the study were corrected.

Reviewer #2: All comments have been addressed properly.This revised version addresses the relevant issues and improves the model setting appropriately. In particular, the sensitivity tests added in the paper are convincing and provide direct answers and improvements to the questions I raised.

7. PLOS authors have the option to publish the peer review history of their article (what does this mean?). If published, this will include your full peer review and any attached files.

Reviewer #1: No

Reviewer #2: No

---

## [Editor Report · Acceptance letter]

5 Apr 2021

PONE-D-20-33893R1 

Global economic impacts of COVID-19 lockdown measures stand out in high-frequency shipping data 

Dear Dr. Verschuur:

I'm pleased to inform you that your manuscript has been deemed suitable for publication in PLOS ONE. Congratulations! Your manuscript is now with our production department. 

Kind regards, 

on behalf of

Professor Bing Xue 

Academic Editor

PLOS ONE